# In Vitro Activity of Cefaclor/Beta-Lactamases Inhibitors (Clavulanic Acid and Sulbactam) Combination Against Extended-Spectrum Beta-Lactamase Producing Uropathogenic *E. coli*

**DOI:** 10.3390/antibiotics14060603

**Published:** 2025-06-13

**Authors:** Ali Atoom, Bayan Alzubi, Dana Barakat, Rana Abu-Gheyab, Dalia Ismail-Agha, Awatef Al-Kaabneh, Nawfal Numan

**Affiliations:** 1Department of Medical Laboratory Sciences, Faculty of Allied Medical Sciences, Al-Ahliyya Amman University, Amman 19328, Jordan; bayann2000@gmail.com (B.A.); danabarakat46@gmail.com (D.B.); ranaabugheyab@gmail.com (R.A.-G.); daloubaidy@yahoo.com (D.I.-A.); 2Pharmacological and Diagnostic Research Center (PDRC), Al-Ahliyya Amman University, Amman 19328, Jordan; 3Princess Iman Center for Research and Laboratory Sciences, Jordanian Royal Medical Services, Amman 11855, Jordan; awatef.kaabneh@jrms.gov.jo; 4College of Pharmacy, The University of Mashreq, Baghdad 10023, Iraq; nawfal.am.numan@uom.edu.iq

**Keywords:** beta-lactam antibiotics, beta-lactamase inhibitors, extended-spectrum beta-lactamase, cefaclor, *Escherichia coli*, clavulanic acid, sulbactam, urinary tract infections

## Abstract

**Background:** Urinary tract infections (UTIs) caused by the multidrug resistance (MDR) phenotype termed extended-spectrum beta lactamase (ESBL)-producing *E. coli* is a significant and growing global health concern. In response to the rising prevalence, the novel Beta Lactam-Beta Lactamase inhibitor (BL/BLI) combinations have been introduced in recent years. While these agents have shown efficacy, their clinical utility is constrained by high cost, limited availability, and emerging resistance mechanisms. The rational of this study was to test the in vitro activity of a cost-effective alternative to currently available BL–BLI combinations against ESBL-producing *E. coli* isolated from urinary tract infections (UTIs). **Objective:** This study investigates the in vitro antimicrobial activity of cefaclor (CFC), both as monotherapy and in combination with the β-lactamase inhibitors clavulanic acid (CA) and sulbactam (SUL), against 52 ESBL-producing *E. coli* isolates derived from urine cultures of patients diagnosed with UTIs. **Methods:** The susceptibility ranges were measured by disk diffusion and minimal inhibitory concentration (MIC) methods. In addition, the Time kill assay and disk approximation method were performed to measure the synergistic and bactericidal activity of the approached combination. **Results:** The MIC50 and MIC90 for CFC were improved from more than 128 µg/mL to 8/4 µg/mL when CFC was combined with either CA or SUL. The triple combination format of CFC/CA/SUL showed MIC50 and MIC90 values at 8/4/4 µg/mL and 64/32/32 µg/mL, respectively. The recovered susceptibility percentages were 54%, 54%, and 58% for CFC/CA, CFC/SUL, and CFC/CA/SUL combinations, respectively. Disk approximation and time–kill assay results revealed synergy and bactericidal effects when CFC combined with CA or SUL for isolates that showed susceptibility restorations of CFC when coupled with CA or SUL by the disk diffusion and MIC method. **Conclusions:** This study proposes a cost-effective combination that could mitigate resistance development and offer a sparing option to last resort treatment choices including carbapenems. However, testing efficacy in a clinical setting is crucial.

## 1. Introduction

*Escherichia coli* (*E. coli*) is the most common member of the *Enterobacteriaceae* family that causes a wide range of human infections, including urinary tract infections (UTIs), gastrointestinal infections, and bloodstream infections. One of the biggest challenges to public health is the rapid development of antimicrobial resistance (AMR) due to the uncontrolled use of broad-spectrum antimicrobial agents (AMAs). Higher resistance rate to AMAs is not only associated with severe infections, but also may lead to critical complications, prolong hospital stay, and increased mortality rates. The World Health Organization (WHO) has listed MDR *E. coli* as one of the organisms that pose a public health threat and require immediate action for therapy development [1].

The extended-spectrum beta-lactamases (ESBL) producing strains of *E. coli* are one of the most prevalent multidrug-resistant (MDR) phenotypes that are associated with complicated infections and limited treatment options [2,3]. More than 400 variants of TEM-1 and SHV-1, along with acquiring CTX-M encoded genes, are the common sources of beta-lactamase enzyme production in ESBL-producing bacteria [4]. The enzyme group that confirms ESBL-producing phenotype comes under class A and D beta-lactamase enzymes in an Ambler classification setup [5,6].

The enzymes produced by ESBL phenotype are capable of hydrolyzing most beta-lactam (BL) agents of cephalosporins and monobactams, which are considered the empirical treatments for most bacterial infections at all body sites. Drug tolerance to other AMAs rather than beta-lactam is additionally standard in ESBL-producing bacteria, leading to limited treatment options. Carbapenems, such as meropenem and imipenem, are considered the gold standard treatments for Gram-negative, ESBL-producing bacteria-associated infections [7]. However, carbapenem overuse may contribute to the accumulation of resistance and lead to the development of carbapenemase-producing organisms such as *Pseudomonas* species, *Acinetobacter* species, and *Stenotrophomonas maltophilia* [8].

The enzymes produced by ESBL can be inhibited by agents termed beta-lactamase inhibitors (BLIs), such as clavulanic acid (CA), sulbactam (SUL), and tazobactam (TZB), which restore BL agent activity in the type of resistance driven by an ESBL phenotype through synergistic activity [9]. BL–BLI combinations are considered an approach to inhibit ESBL and serve as an option to spare carbapenem and other last-resort AMAs. This was supported by previous reports, which showed that using a piperacillin/tazobactam combination significantly reduced the prevalence of ESBL-associated infections [10,11]. The available oral BL–BLI combinations effective against ESBL infections are currently very limited. Previous investigations have observed promising results for treating bacteria harboring ESBL phenotypes through utilizing different BL–BLI combinations [12,13,14,15]. However, an optimum BL–BLI blend remains to be determined.

In this study, an approach of cefaclor (CFC) in combination with CA or SUL was analyzed against multiple uropathogenic ESBL isolates to provide treatment options that could be tested in clinical settings. Cefaclor (CFC) is a second-generation oral cephalosporin that inhibits bacterial cell wall synthesis and is effective against a range of Gram-positive and Gram-negative organisms. While it can be used to treat skin and urinary tract infections, it is primarily prescribed for respiratory tract infections due to its proven efficacy against common respiratory pathogens such as *Streptococcus pneumoniae*, *Haemophilus influenzae*, and *Moraxella catarrhalis* [16,17,18]. Cefaclor demonstrates superior mucosal tissue penetration and has higher oral bioavailability compared to amoxicillin/clavulanic acid and other oral cephalosporins. Its rapid absorption and renal excretion contribute to high tissue concentrations especially at the urinary system and are associated with a lower incidence of gastrointestinal side effects [18,19]. On the other hand, CA and SUL were selected because these BLIs exist in different combinations, well known for their use to inhibit a wide range of beta-lactamase enzymes specifically produced by ESBL isolates, besides their availability and cost affordability.

We believe that an optimum beta lactamase inhibitor combined with CFC could provide an optimum treatment option. Despite utilizing a standard technique, the activity of the selected combinations in this study needs clinical trials to support the applicability of the in vitro findings.

## 2. Results

### 2.1. Antibiogram Susceptibility Analysis of Selected Isolates

A total of 52 *E. coli* clinical isolates with conclusive ESBL production were selected in this study. The analysis of the susceptibility testing for the selected isolates indicated a high resistance rate to the commonly used cephalosporin drugs as a first-line treatment option. This was noticed after observing a prominent resistance rate to the AMAs cefuroxime, ceftazidime, ceftriaxone, cefepime, ciprofloxacin, and cefazolin ranging from 72 to 100% (Table 1). In contrast, these isolates showed high susceptibility rates to carbapenem agents, including ertapenem, imipenem, and meropenem, along with amikacin, gentamycin, nitrofurantoin, and fosfomycin with susceptibility rates ranging from 76 to 100% (Table 1). The susceptibility rate to the AMAs combinations Amoxicillin/Clavulanic acid, and Piperacillin/Tazobactam were 28% and 64%, respectively.

### 2.2. Analysis of CFC Combination with Selected Beta Lactamases Inhibitors

#### 2.2.1. Disk Diffusion Method

The disk diffusion method was utilized at the beginning to screen if the suggested combination could improve the inhibition zone and restore the susceptibility of CFC against the isolates collected. The inhibition zones diameter generated around disks containing cefaclor alone or in combination with CA and/or SUL were recorded and analyzed for susceptibility restoration of CFC. The mean of the inhibition zone diameters of CFC alone for ESBL-producing isolates (total 52 isolates) was 6.6 mm. The mean of inhibition zones was improved to 16.4, 18.3, and 19.6 mm for CFC/CA, CFC/SUL, and CFC/CA/SUL combinations, respectively (Table 2). These improvements restored the susceptibility of CFC to 54%, 56%, and 69.2% for CFC/CA, CFC/SUL, and CFC/CA/SUL, respectively (Table 2).

Disks containing CA, SUL, or CA/SUL were used as controls, and minimal effects on the inhibition zone were observed. CFC susceptibility breakpoints from the CLSI M100 guidelines for 2023 indicate susceptibility for inhibition zone diameters equal to or higher than 18 mm, intermediate for 15–17 mm, and resistance for equal to or lower than 14 mm [20].

#### 2.2.2. MIC Determination of the Combination Strategies Using Broth Microdilution Method

The standard microdilution method was performed to measure the activity of selected combinations against *E. coli* isolates to confirm the result obtained by disk diffusion. The result analysis revealed that the mean of MIC_50_ and MIC_90_ and the restoration of susceptibility to CFC were highly improved to relatively comparable results obtained from the disk diffusion method (Table 3). The MIC_50_ and MIC_90_ for CFC were improved from more than 128 µg/mL to 8/4 µg/mL when CFC was combined with either CA or SUL. The triple combination format of CFC/CA/SUL showed MIC_50_ and MIC_90_ values at 8/4/4 µg/mL and 64/32/32 µg/mL, respectively. CFC recovered the susceptibility percentages to 54%, 54%, and 58% for CFC/CA, CFC/SUL, and CFC/CA/SUL combinations, respectively (Table 3). All combination approaches generated comparable MIC values and restoration percentages.

#### 2.2.3. Synergistic Analysis of Selected Combination by Disk Approximation Method

The disk approximation method was utilized to gain more insights into the synergistic activity between agents in the selected combinations. To check and confirm this activity, the disk approximation method was performed (Figure 1). Our result indicated an increased growth clearance between the adjacent disks for isolates that showed susceptibility restorations of CFC when coupled with CA or SUL by the disk diffusion and MIC method (Table 4). All isolates (n = 23) that observed resistance to the selected combinations also showed no growth inhibition between adjacent disks holding CFC, CA, and SUL. The data indicated a compatibility between previously used methods to test the selected combinations. In addition, CA and SUL act by synergy mechanism when combined with CFC.

#### 2.2.4. Time Kill Assay Results for Synergy Examination

Accordingly, time–kill assays were performed to confirm the results obtained by the disk approximation method and to obtain more insight into the killing (bactericidal or bacteriostatic) and synergistic activities of CFC of different combination formats. This assay was performed at the MIC_50_ points determined in the previous section for ESBL isolates (8/4/4 µg/mL). It was the most frequent points observed for ESBL isolates showing restoration to the susceptibility range. In addition, the concentration of 8 µg/mL of CFC is the highest value at the breakpoint of the susceptibility range of CFC according to CLSI M100 guidelines for 2023 [20]. Figure 2 shows the time–kill curves for four randomly selected ESBL isolates. The results indicated that the CFC combinations with CA, SUL, or CA/SUL reduced the log_10_ CFC/mL for more than 3 log_10_ compared with CFC alone against ESBL isolates that observed susceptibility restoration by the MIC method. In contrast, ESBL isolates deficient for restoration by the MIC method did not observe the required logarithmic reduction in CFC/mL for all possible combinations.

These data indicated that the combinations CFC/CA, CFC/SUL, and CFC/CA/SUL were bactericidal and acted upon by synergistic effects for ESBL isolates that show susceptibility restoration at concentrations equal to 8, 4, and 4 µg/mL for CFC, CA, and SUL, respectively, with comparable results for all combination formats.

## 3. Discussion

As a result of the lack of development of new antimicrobial agents, rejuvenating existing AMAs in a combination format is one possible strategy to treat multidrug-resistant associated infections. Antibiotic combinations expand the spectrum of activity and possibly delay the emergence of resistance [21]. This study tested the in vitro efficacy of CFC combinations with different BLIs, including CA and SUL, against clinically collected ESBL isolates of *E. coli* as the leading cause of UTI worldwide [22].

The susceptibility analysis of the collected isolates indicated that ESBL isolates are highly resistant to most cephalosporines included in the antibiogram list, which confirms the ESBL resistance pattern. On the other hand, the isolates were susceptible to piperacillin-tazobactam, carbapenems, amikacin, gentamycin, nitrofurantoin, and fosfomycin to various degrees. However, these treatment options are not listed in the first-line treatment, are associated with severe side effects, have limited therapeutic dose achievement in the urinary system, especially in the case of a complicated UTI, and have boundaries associated with availability, route of administration, and cost especially for low-income countries [7,23].

In this study, we found an improvement in the susceptibility range of CFC when combined with CA or SUL for more than 50% of the tested ESBL isolates. The restored susceptibility of the combined agents was bactericidal through a synergy mechanism, as seen by the time–kill assay. However, the CFC combination with CA or SUL did not restore the susceptibility pattern of CFC to all strains with a confirmed ESBL phenotype. One explanation is that these strains are co-producers of other enzymes such as Ambler class C, Metallo-beta-lactamase, AmpC, and K1 that resist neutralization by BLI, interfere with ESBL phenotypic detection, are not inhibited by BLI, and harbor a wide range of resistance. Also, these strains could carry mutations at the binding site of CFC (modified penicillin-binding protein mutations), harbor drug efflux pumps, or possess a change in membrane permeability due to the loss of porins [24,25,26,27]. However, genetic analysis to determine the molecular elements associated with a failure of the approached combination, and the exact mechanisms of associated resistance can help to propose more targeted BL–BLI combinations.

Compared with other combinations, the tested isolates showed a 28% and 64% susceptibility rate to amoxicillin/CA and piperacillin/TZB combination, respectively (Table 1). This indicated that the selected combinations of CFC with CA or SUL are considerably better than that of amoxicillin/CA, but not the piperacillin/TZB combination. However, several studies have witnessed discrepancies between the in vitro susceptibility data of piperacillin/TZB and the clinical usefulness of this combination, as there are better outcomes in patients with very low MIC value and low inoculum UTI and bloodstream bacterial infections [28,29,30]. The clinical outcome of using piperacillin/TZB is improved after modifying the breakpoint interpretation criteria for this agent [31].

To the best of our knowledge, no earlier literature has reported the use of the CFC/BLI combination. However, comparisons can be made with other combination approaches involving cephalosporins/BLI pairings. In one report, the AMAs cefpodoxime, ceftibuten, cefixime, and cefdinir were tested in vitro against collected ESBL isolates in combination with CA, and restoration of susceptibility was observed for these agents ranging from 21 to 49% with a ceftibuten/CA combination having the best restoration result [15]. Consequently, TZB along with SUL combinations with different cephalosporins were also investigated, and an enhancement of susceptibility was observed ranging to more than 60% compared to each cephalosporin with cefepime/TZB showing the highest activity (100% restoration rate) against ESBL producer isolates [32]. To this point, all previous BL–BLI-based combinations studies observed quite similar results to our selected combination, with the cefepime/TZB combination having the greatest susceptibility restoration rate. Recently, several studies have investigated the in vivo and in vitro efficacy of a ceftibuten/CA combination and observed promising efficacy in managing UTIs caused by ESBL producers [33,34,35]. These studies highlighted the reproducibility of a CA-based combination that required perfectly matched BL agents such as ceftibuten. In early reports published between 2008 and 2009, CA showed more than 80% susceptibility range to ESBL producer Gram-negative isolates, including *E. coli*, when coupled with different cephalosporins such as cefdinir, cefpodoxime, or cefixime [36,37]. Overall, the susceptibility rates to different BL–BLI combinations are decreasing over time. It seems that the evolution of beta-lactamase enzymes is the main mechanism of resistance driven by pathogenic bacteria against BL antibiotics.

This report did not find a significant difference between a CA and/or SUL combination with CFC regarding susceptibility restoration rate. However, previous reports have shown that increasing the SUL ratio in combination with cefoperazone/SUL improved the susceptibility ranges of cefoperazone against isolates of ESBL *E. coli* [38,39]. It is worth mentioning that there were no clinical restrictions for our combination approach concerning pharmacokinetics and dynamics at the concentration tested [40]. In another report, TZB was more effective than SUL or CA in different approaches of BL–BLI-based combinations [32,41]. Novel BLIs, including ETX0282, ETX1317, VNRX7145, and VNRX5236, have been coupled with cefpodoxime or ceftibuten individually and demonstrated promising results against ESBL isolates, where these novel BLIs are still under investigation [34]. All these findings highlighted the importance of BLI type and concentration in obtaining better antimicrobial activity.

The disk approximation method used in this study confirmed CFC’s restoration and synergistic effects after CA/SUL combinations by spotting growth inhibition between adjacent disks loaded with each drug individually. All samples with restored susceptibility also showed positive synergy in the disk approximation method. These highlighted the feasibility of this method for detecting synergy for any proposed BL–BLI combination. One study showed that a cefixime and amoxicillin-clavulanate combination had a positive in vitro synergy by the disk approximation method and a complete in vivo resolution of patients infected with ESBL *E. coli* after completion of the treatment course [14]. Furthermore, a previous study showed that the disk approximation method can detect synergistic activity between cefpodoxime or cefdinir with amoxicillin-clavulanate against clinical isolates of *Enterobacteriaceae* with CTX-M, SHV, or TEM ESBLs phenotypes [13]. All things considered, an updated antibiogram list containing a BL–BLI combination is an approach to be performed routinely in a clinical setting using the simple disk approximation method.

In clinical setting this study observed potential data of a second-generation cephalosporin CFC that can be rejuvenated in combination with CA or SUL. Such a combination approach is an oral treatment option that is cost-effective and can be safely prescribed for outpatients suffering from cystitis due to ESBL uropathogenic strains. Of greatest relevance, multiple in vivo reports evaluated CFC treatment for patients with uncomplicated UTIs and observe a success rate between 70 and 94%, with more than 50% for *E. coli*-ESBL infections. The observed excellent efficacy and tolerability were superior to other cephalosporins. These findings were attributed to the bioavailability of CFC at the urinary system as the principal rout of CFC excretion [19,42]. Despite some limitations in this study which include limited isolates number, genotypic profiling, pharmacokinetic analysis, and in vivo testing at clinical setting, this combination could offer a sparing choice to carbapenems, especially for pregnant women and patients who cannot tolerate carbapenem treatments.

## 4. Materials and Methods

### 4.1. Isolates Collection

*E. coli* isolates with ESBL-positive and carbapenemase-negative results were collected from September to December 2023 from Princess Iman Research Center in Amman, Jordan. All isolates were obtained from the positive urine cultures of patient urine samples sent to the lab for urine culture and antimicrobial susceptibility testing. The identification and susceptibility testing were performed using the VITEK-II automated system (BioMérieux, Craponne, France). Isolates were further confirmed upon delivery to our lab by conventional biochemical testing and cultivating on Eosin Methylene Blue (EMB) agar plates (Condalab, Madrid, Spain). A total of 52 culture-confirmed *E. coli* isolates were selected after excluding any duplicate isolates obtained from repeated culture with similar patient ID or any isolates with inconclusive phenotypic testing for ESBL. Isolates were stored in nutrient broth containing 20% glycerol and kept at −70 °C until analysis time.

### 4.2. Phenotypic Confirmatory Testing for ESBL Production by Combination Disk Test (CDT)

For ESBL testing, a 0.5 McFarland standard turbidity was prepared for each isolate, and then the plates were swabbed onto Mueller–Hinton agar (MHA; Oxoid, Hampshire, UK). After that, 30 μg of ceftazidime and cefotaxime were applied to the disks alone and in combination with 10 μg of clavulanic acid (Mast Group, Bootle, UK). The inhibition zones were measured following an incubation period at 35 ± 2 °C overnight. A positive ESBL test is indicated by an inhibition zone diameter (including the disk itself) measuring ≥ 5 mm with CA compared to the antibiotic alone, according to the Clinical and Laboratory Standards Institute (CLSI) M100 recommendations from 2023 [20].

### 4.3. Antimicrobial Susceptibility Testing of the Proposed Combination

#### 4.3.1. Antimicrobial Agent Preparations

All AMAs included in this study were prepared according to CLSI guidelines for AMA preparation criteria listed in the CLSI M100 guidelines from 2023 [20]. The following formula was used to determine each antibiotic’s potency based on the supplier’s provided water content, Purity, and active fraction:Potency = Purity × Active fraction × (1 − water content)

The volume needed to prepare antimicrobial stock solutions was calculated using the following equation:(Volume (mL) = mass (mg) × Potency (μg/mg))**/**Concentration (μg/mL).

ATCC 25922 *E. coli* strain was used as the quality control isolate. The susceptibility pattern of this isolate, as recorded on the data sheet, was compared with the results obtained for the prepared antimicrobial agents and found to be consistent.

#### 4.3.2. Kirby–Bauer Disk Diffusion Method

The following disk contents were used for this method: 30 µg for CFC, 10 µg for CA, and 10 µg for SUL. These contents were obtained from the CLSI M100 guidelines mimicking other combination formats that contain CA or SUL [20]. All AMAs, individually or in combination, were loaded on blank disks (Oxiod, Hampshire, UK). Subsequently, these disks were applied onto MHA plates streaked with 0.5 McFarland standard inoculum of individual bacterial isolate and then incubated overnight at 35 ± 2 °C. The diameter of the inhibition zone was measured in millimeters (mm) and recorded for further data analysis.

#### 4.3.3. Broth Microdilution Assay

Using a 96-well microtiter plate, a double-fold serial dilution of CFC, either individually or in different possible formats with CA and SUL, was prepared at a volume equal to 100 µL before adding the bacterial inoculum. The initial concentrations of AMAs were 128 µg/mL for CFC and 64 µg/mL for the BLI (CA and SUL) at a ratio of 2:1 after the addition of the Mueller–Hinton broth (MHB; Oxoid, Hampshire, UK) containing 0.5 McFarland adjusted turbidity of each isolate to each well in the prepared microtiter plate and incubated overnight at 35 ± 2 °C. The last well with the lowest antibiotic concentration that did not exhibit any visible growth was recorded as the MIC point. MIC_50_ and MIC_90_ determination points were performed as described previously [43].

#### 4.3.4. Disk Approximation Method

This test is designed to detect the potential synergistic effect of two different antimicrobial agents when placed in proximity of 15 to 20 mm center-to-center in a disk diffusion assay [44]. Blank disks were loaded with CFC, CA, or SUL, placed on agar plates, and incubated overnight at 35 ± 2 °C after the plates had been inoculated with individual bacterial isolate adjusted to 0.5 McFarland standard turbidity.

The clearance of growth in between adjacent disks was observed and reported as synergy if there was a cleared growth area between the two disks applied, partial synergy if the area between the two disks was inhibited but not cleared, or no synergy if there was minimal to no inhibition between the applied disks.

#### 4.3.5. Time–Kill Assay

A time–kill assay is considered a robust technique to evaluate the bactericidal or bacteriostatic effect of antimicrobial agents under testing. Also, it can be used to detect synergy or antagonism effects of suggested combinations by comparing the level of growth inhibition at different time points of combined agents vs. individual drugs within 24 h. A bactericidal effect can be considered if an agent decreases the CFU of bacterial growth more than or equal to 3 log_10_ within 24 h compared to the initial time point. On the other hand, the synergy effect of combined agents should inhibit the growth of bacteria at a level more than or equal to 2 log_10_ CFU/mL when compared with the most active individual agent in the suggested combination [44].

Time kill assays were performed according to the procedures outlined in CLSI M26-A guidelines [45]. In separate 10 mL tubes, bacterial suspension of each randomly selected isolates were incubated with CFC alone or CFC/CA, CFC/SUL, and CFC/CA/SUL combinations at concentrations equal to the MIC determined in the broth microdilution method. A diluted fraction equal to 0.1 mL was taken from each tube at different time intervals of 0, 4, 8, 12, and 24 h, inoculated on sterile MHA, followed by overnight incubation at 35 ± 2 °C. A colony count was then performed, and the log10 value for each combination formatted at different time intervals was calculated.

#### 4.3.6. Data Analysis

The percentages of susceptibility, intermediate, and resistance to each AMA were calculated by dividing the number of susceptible, intermediate, or resistance by the total number of ESBL isolates. To calculate the MIC50 and MIC90, the MIC values obtained from the microtitration method were ordered from the highest to the lowest. Then, values at the 50%, and 90% positions were considered the MIC50 and MIC90 points, respectively. The rate of CFC restoration for each combination format was calculated by dividing the number of isolates that revert susceptible after CFC/BLI treatment by the total number of isolates tested. The time–kill assay curve was generated by plotting the incubation time intervals on the *X*-axis vs. the mean of log10 CFU for individual agents and combination regimens on the *Y*-axis. All calculations and graph presentations were performed using Microsoft Excel software (Microsoft Corp., Redmond, WA, USA).

## 5. Conclusions

There is a serious demand to investigate novel, effective therapies for MDR pathogens. Rejuvenating old antibiotics in an adjuvant combination approach such as BLI can mitigate resistance development and provide a sparing option to last-resort AMAs used to treat complicated MDR-associated infections [21,40,46]. This study’s proposed combination is cost-effective. We suggested including CFC/CA or CFC/SUL combination in the antibiogram list for uncomplicated UTIs to obtain conclusive statistical data on the susceptibility pattern of these combinations. Moreover, evaluating both microbiological response and clinical treatment outcomes in patients with UTI is essential. This combination approach can serve as empirical treatment for uncomplicated UTI due to its bioavailability and spectrum of activity against Gram-negative bacteria. Further studies are needed to optimize methodologies that predict BL–BLI combination efficacy in a clinical setting, as they can act as good substitutes for carbapenems.

## Figures and Tables

**Figure 1 antibiotics-14-00603-f001:**
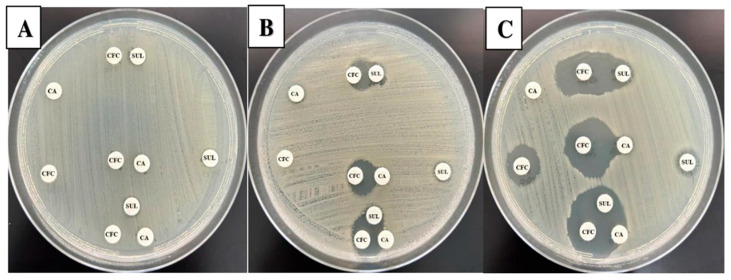
Disk approximation method. The figure displays the results for three selected isolates. In each plate (labeled **A**–**C**), disks holding CFC, CA, and SUL were applied separately or adjacent to each other with the possibilities of CFC:CA, CFC:SUL, and CFC:CA:SUL. Panel (**A**) represents an isolate that showed resistance to all individual agents and lack of synergy between all combined possibilities. On the other hand, panels (**B**,**C**) revealed partial and full synergy between all adjacent agent possibilities.

**Figure 2 antibiotics-14-00603-f002:**
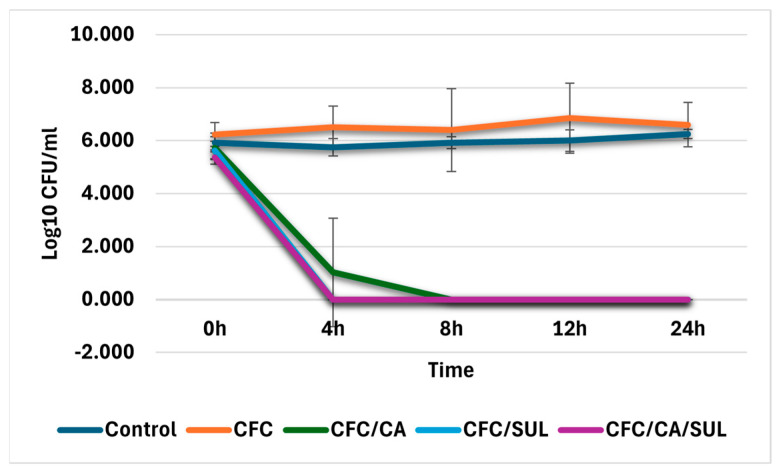
Time–kill assay results. The Log10 CFU/mL was plotted against different times of incubations. Four representative isolates were analyzed, and the blue line indicates bacterial growth control. Error bars represent the mean of log 10 CFC/mL of two independent experimental repeats. Abbreviations: CFC—cefaclor, CA—clavulanic acid, SUL—Sulbactam.

**Table 1 antibiotics-14-00603-t001:** Susceptibility Testing Analysis of Collected *E. coli* Isolates.

Antimicrobial Agent	Total Isolates (n = 52)			
	S%	I%	R%	SDD%
Amoxicillin/Clavulanic acid	28	22	50	-
Piperacillin/Tazobactam	64	8	28	-
Cefuroxime	-	-	100	-
Ceftazidime	11	8	81	-
Ceftriaxone	6	-	94	-
Cefepime	14	-	72	14
Ertapenem	100	-	-	-
Imipenem	97	-	3	-
Meropenem	97	-	3	-
Amikacin	100	-	-	-
Gentamicin	67	-	33	-
Ciprofloxacin	11	17	72	-
Trimethoprim/Sulfamethoxazole	25	-	75	-
Cefazolin	-	-	100	-
Nitrofurantoin	76	10	14	-
Fosfomycin	92	-	8	-

Data on the table is the susceptibility rates calculations for 52 *E. coli* isolates obtained from positive urine culture. Patients’ susceptibility profiles were generated using the VITEK-II automated system. Abbreviations: S—susceptible, I—Intermediate, R—Resistant, SDD—susceptible dose dependent.

**Table 2 antibiotics-14-00603-t002:** Data Analysis of Inhibition Zone Measurements for Individual Agents and Combination Formats Using Disk Diffusion Assay.

AMAs	Total Isolates (n = 52)		
	Inhibition Zone Range (mm) *	Mean of Inhibition Zone (mm) †	Percent of Susceptibility Restoration (Number of Restored Isolates) ‡
**CFC**	0–18	6.6	-
**CFC/CA**	0–27	16.4	54.0 (28)
**CFC/SUL**	0–28	18.3	56.0 (29)
**CFC/CA/SUL**	0–30	19.6	69.2 (36)
**CA**	0–12	1.7	0 (0)
**SUL**	0–14	8.7	0 (0)
**CA/SUL**	0–14	11.9	0 (0)

* Data are the highest and lowest zone diameters measurements in mm for all tested isolates. † Data represent the mean of zones diameter measurements in mm, and calculated by obtaining the summing of all diameters measurements divided by the number of isolates tested. ‡ Data represent the percentages of restoration relative to CFC and calculated by dividing the number of isolates that revert susceptible after CFC/BLI treatment by the total number of isolates tested. Abbreviations: CFC—cefaclor, CA—clavulanic acid, SUL—Sulbactam. Data are representative of two independent experiments.

**Table 3 antibiotics-14-00603-t003:** Data Analysis of MIC Results Using Microtitration Assay.

AMAs	Total Isolates (n = 52)			
	MIC Range (µg/mL) *	MIC_50_ (µg/mL) †	MIC_90_ (µg/mL) †	% of CFC Susceptibility Restoration (Number of Restored Isolates) ‡
**CFC**	8–>128	>128	>128	-
**CFC/CA**	4/2–128/64	8/4	128/64	56.0 (29)
**CFC/SUL**	8/4–128/64	8/4	128/64	56.0 (29)
**CFC/CA/SUL**	4/2–128/64	8/4/4	64/32/32	58.0 (30)
**CA/SUL**	>64/>64	>64/>64	>64/>64	0.0 (0)
**CA**	>64	>64	>64	0.0 (0)
**SUL**	>64	>64	>64	0.0 (0)

* Data stands for the highest and lowest MIC values obtained from all tested isolates. † The MIC values obtained from the microdilution method were arranged in descending order. The values corresponding to the 50% and 90% position were identified as the MIC_50_ and MIC_90_, respectively. ‡ Data represent the percentages of restoration relative to CFC and calculated by dividing the number of isolates that revert susceptible after CFC/BLI treatment by the total number of isolates tested. Abbreviations: CFC—cefaclor, CA—clavulanic acid, SUL—Sulbactam. Data are representative of two independent experiments.

**Table 4 antibiotics-14-00603-t004:** Disk Approximation Method Results.

Interpretation Criteria	CFC/CA	CFC/SUL	CFC/CA/SUL
No synergy	23	23	22
Partial synergy	3	3	2
Full synergy	26	26	28

The numbers on the table represent the total number of isolates that observed no, partial, or full synergy category for each combination format. Abbreviations: CFC—cefaclor, CA—clavulanic acid, SUL—Sulbactam. Data are representative of two independent experiments.

## Data Availability

The data obtainable in this study are enclosed within this article.

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
