# Peer review of "In Vitro Activity of Cefaclor/Beta-Lactamases Inhibitors (Clavulanic Acid and Sulbactam) Combination Against Extended-Spectrum Beta-Lactamase Producing Uropathogenic *E. coli"

_antibiotics, 2025, doi:10.3390/antibiotics14060603_

Round 1

Reviewer 1 Report

Comments and Suggestions for Authors

The manuscript titled “In vitro Activity of Cefaclor/Beta-Lactamase Inhibitors (Clavulanic Acid and Sulbactam) Combination against Extended-Spectrum Beta-Lactamase Producing Uropathogenic E. coli” by Atoom et al. is important and worthy of publication because it addresses the urgent need for alternative oral treatment options against ESBL-producing E. coli which is a major UTI pathogen

15-17: “in vitro activity…” is vague. What is the rational/hypothesis behind the study?

18: Italicize E. coli

29-30: The conclusion is speculative. Please qualify how the study will achieve low risk of generating bacterial resistance. Also, strengthen it with call for clinical studies/validation

38: Enterobacterales is an order not a family

Introduction

The Introduction section should be rearranged to flow thus: AMR background, ESBL challenge, treatment limitations, and rationale for cefaclor combination with BLI.

The knowledge gap should be emphasized, by mentioning whether few or no study have assessed the efficacy of cefaclor or other second-generation cephalosporins with BLIs against ESBL-producing strains

Redundant facts like like cefaloclor’s use in penicillin-allergy or pregnant patients should be removed

38–44: There is repetition of general information on AMR and WHO guidance

75–81: Please make more concise of cefaclor’s clinical uses

Methods

Which statistical analysis was used to evaluate significance? There should be a Statical Analysis section where the method and uses of the statistical package is described

Although the recommendation of a more robust ESBL detection was mentioned in 277-278, molecular tests to confirm ESBL genes are important in buttressing the mechanistic insight

349–359: The isolate collection method is clear, but strain selection criteria (e.g., duplicate isolates, exclusion criteria) are missing.

373–403: Reference CLSI/EUCAST standards clearly

Clearly state whether the test was performed in triplicate or duplicate to ensure result reliability

Results

131-136: Repetition of justification and methodology

142-145: Better suited for the discussion section

Table 1 to Table 4: Undefined abbreviations

What is the unit of measurements in table 4? The title of table 4 should not be the same with the heading of the first column “Disc approximation result”

Figure 1 was not referred to in the text

The data in the tables 1 to 3 are heavy and not well summarized

202-213: Better suited in the justification/rationale for methodology

The heading inside the figure 2 should be removed

218-219: CLSI criteria should be in the Methodology. Why was CLSI 2025 criteria not used?

Discussion

The discussion should be made more compact

There is limited interpretation of results’ implications for clinical practice.

264–273: Resistance mechanisms were repeated without connecting them to the study’s isolates

279–322: Although many previous works were cited, they were not clearly related to the current findings. Authors should focus on comparing their findings with 2 to 3 major relevant studies.

They should also highlight potential reasons for partial resistance recovery

The  limitations of the study should be discussed explicitly (no genotypic profiling and pharmacokinetics, and small sample size)

Conclusion

The conclusion is weak as it restates justification of study and results without clear clinical translation

435: Proposing animal use is speculative and unsupported by data. Authors should replace speculation with focused implications and emphasize clinical caution and call for in vivo testing.

Reviewer 2 Report

Comments and Suggestions for Authors

The present manuscript evaluated the in vitro efficacy of cefaclor (CFC), alone and in combination with the β-lactamase inhibitors clavulanic acid (CA) and sulbactam (SUL), against ESBL-producing E. coli isolates from urine samples. The proposed combinations are cost-effective and potential options for treating multidrug-resistant infections, with possible applications in both human and veterinary medicine. However, the Methods section requires significant improvements for clarity, reproducibility, and scientific rigor. I recommend major revision. Please address the following points:

  1. Table 1: Include a dedicated Methods subsection describing: The concentration of antibiotics used in susceptibility testing (e.g., disk content or MIC range); the interpretive criteria used (e.g., CLSI M100 guidelines); the testing method applied (e.g., disk diffusion, broth microdilution, etc.) Additionally, the legend should clearly state the method used.
  2. Tables 2 and 3: Include in the table legends that the reported values represent the mean ± standard deviation (or just mean) of n biological replicates. Specify the number of independent repeats used to derive the values.
  3. Table 4: Clarify what the numbers in the table represent (e.g., number of isolates etc.). Make the table headings and legend self-explanatory, so the table can be understood without referring back to the text.
  4. Include details of controls used for each assay in the method section.
  5. Provide a brief statistical analysis section stating the number of replicates per experiment, any statistical tests used.
  6. Line 369: Clarify: does "≥ 5 mm" refer to the radius or diameter of the inhibition zone? Does the measurement include the disc itself? State the diameter of the discs used.

Provide sufficient methodological detail to allow replication of this assay.

  1. Provide appropriate references for all experimental protocols, especially for synergy determination using the Disc Approximation Method (e.g., standard sources or peer-reviewed literature).
  2. Line 137 and throughout: Clearly specify whether values refer to the radius or diameter of the zone of inhibition. Indicate the number of replicates used to calculate the mean. Revise for consistency across the manuscript. For example:

“The mean diameter of the inhibition zone for CFC alone against 52 ESBL-producing isolates was 6.6 mm…”

Round 2

Reviewer 1 Report

Comments and Suggestions for Authors

The authors have carefully responded to my comments which improved the manuscript for publication

Reviewer 2 Report

Comments and Suggestions for Authors

The authors have addressed all the queries